# Peripheral Blood Microbiome Analysis via Noninvasive Prenatal Testing Reveals the Complexity of Circulating Microbial Cell-Free DNA

Xunliang Tong,[a] Xiaowei Yu,[b] Yang Du,[c] Fei Su,[d] Ye Liu,[d] Hexin Li,[d] Yunshan Liu,[c] Kai Mu,[e] Qingsong Liu,[f] Hui Li,[g] Jiansheng Zhu,[h] Hongtao Xu,[i] (ID) Fei Xiao,[d,j] Yanming Li[a]

aDepartment of Pulmonary and Critical Care Medicine, Beijing Hospital, National Center of Gerontology, Institute of Geriatric Medicine, Chinese Academy of Medical Sciences, Beijing, China

bCenter for Reproductive Medicine, Center for Prenatal Diagnosis, First Hospital, Jilin University, Changchun, Jilin, China

cAnnoroad Gene Technology Co., Ltd., Beijing, China

dClinical Biobank, Beijing Hospital, National Center of Gerontology, Institute of Geriatric Medicine, Chinese Academy of Medical Sciences, Beijing, China

eDepartment of Medical Genetics, Zibo Women and Children Hospital, Zibo, China

fChengdu Women's and Children's Central Hospital, School of Medicine, University of Electronic Science and Technology of China, Chengdu, China

gClinical Laboratory, Maternal and Child Health Hospital of Hubei Province, Wuhan, China

hMedical Genetic Center, Maternity and Child Health Hospital of Anhui Province, Hefei, China

iDepartment of Laboratory Medicine, Beijing Hospital, National Center of Gerontology, Institute of Geriatric Medicine, Chinese Academy of Medical Sciences, Beijing, China

jThe Key Laboratory of Geriatrics, Beijing Hospital, National Center of Gerontology, Institute of Geriatric Medicine, Chinese Academy of Medical Sciences, Beijing, China

Xunliang Tong, Xiaowei Yu, and Yang Du are co-first authors. Author order was determined by the corresponding authors after negotiation.

**ABSTRACT**   While circulating cell-free DNA (cfDNA) is becoming a powerful marker for noninvasive identification of infectious pathogens in liquid biopsy specimens, a microbial cfDNA baseline in healthy individuals is urgently needed for the proper interpretation of microbial cfDNA sequencing results in clinical metagenomics. Because noninvasive prenatal testing (NIPT) shares many similarities with the sequencing protocol of metagenomics, we utilized the standard low-pass whole-genome-sequencing-based NIPT to establish a microbial cfDNA baseline in healthy people. Sequencing data from a total of 107,763 peripheral blood samples of healthy pregnant women undergoing NIPT screening were retrospectively collected and reanalyzed for microbiome DNA screening. It was found that more than 95% of exogenous cfDNA was from bacteria, 3% from eukaryotes, and 0.4% from viruses, indicating the gut/environment origins of many microorganisms. Overall and regional abundance patterns were well illustrated, with huge regional diversity and complexity, and unique interspecies and symbiotic relationships were observed for TORCH organisms (*Toxoplasma gondii*, others [*Treponema pallidum* {causing syphilis}, hepatitis B virus {HBV}, and human parvovirus B19 {HPV-B19}], rubella virus, cytomegalovirus [CMV], and herpes simplex virus [HSV]) and another common virus, Epstein-Barr virus (EBV). To sum up, our study revealed the complexity of the baseline circulating microbial cfDNA and showed that microbial cfDNA sequencing results need to be interpreted in a more comprehensive manner.

**IMPORTANCE** While circulating cell-free DNA (cfDNA) has been becoming a powerful marker for noninvasive identification of infectious pathogens in liquid biopsy specimens, a baseline for microbial cfDNA in healthy individuals is urgently needed for the proper interpretation of microbial cfDNA sequencing results in clinical metagenomics. Standard low-pass whole-genome-sequencing-based NIPT shares many similarities with the sequencing protocol for metagenomics and could provide a microbial cfDNA baseline in healthy people; thus, a reference cfDNA data set of the human microbiome was established with sequencing data from a total of 107,763 peripheral blood samples of healthy pregnant

Address correspondence to Fei Xiao, xiaofei3965@bjhmoh.cn, or Yanming Li, liyanming2632@bjhmoh.cn.

The authors declare no conflict of interest.

women undergoing NIPT screening. Our study revealed the complexity of circulating microbial cfDNA and indicated that microbial cfDNA sequencing results need to be interpreted in a more comprehensive manner, especially with regard to geographic patterns and coexistence networks.

**KEYWORDS** NIPT, microbiome, cfDNA, population-based analysis, noninvasive prenatal testing, cell-free circulating DNA

Microbes occupy habitats like the gut, skin, vagina, and other organs of the human body, and these microbiota are thought to be essential for maintaining human health (1–3). In recent years, bacterial and viral nucleic acid fragments have been identified from the circulation of asymptomatic subjects (4–7), which has enabled sequencing of circulating cell-free DNA (cfDNA) from liquid biopsy specimens for noninvasive diagnosis of infectious diseases and improved our understanding of the possible functions of the resident microbiota (8).

Noninvasive prenatal testing (NIPT) using massively parallel sequencing of cfDNA in maternal plasma has recently changed the clinical paradigm of prenatal screening for common fetal autosomal aneuploidies (9). Low-depth sequencing has been found to be a cost-effective strategy for detecting genetic associations with special DNA viruses for complex traits (10), suggesting that NIPT data may be used to establish a baseline for blood microbes in peripheral blood and facilitate the clinical interpretation of infectious diseases.

In our current study, we tried to capture the microbial signatures among peripheral blood samples obtained in a nationwide population-based NIPT study and identify their clinical significance. Based on data from a large set of samples, we constructed a reference data set of peripheral blood microbes and filled in the blanks for background microorganisms in China. Measuring the microbial abundance in each region helped to correct the background values for pathogen identification. We also investigated the bacterial and viral ecologies in human blood, as well as their coexistence and the interactions between these microbes, via population-level coexistence network analysis and conditional differential abundance analysis.

## RESULTS

**Quality control and simulation of potential saturation effect.** Random individual libraries sized from 1 to 2,000 were sampled from the collection. For each sample size, 1,000 sampling runs were done to conclude a boxplot. In total, 1,697 species were detected (with at least 1 uniquely mapped read in all samples), out of 4,939 unique species checked against the human microbiome reference genome (HMREFG) of the Human Microbiome Project (HMP; http://hmpdacc.org/HMREFG/) list. The estimated species counts and the associated standard deviations were illustrated using a box plot for each sample size (Fig. 1A), and no saturation effect was found.

**Subjects and general information.** The 107,763 participants were recruited from 28 provinces, autonomous regions, and municipalities in mainland China. Inner Mongolia, Tibet, and Qinghai were not included in this study due to lack of sufficient samples. Among all the province-level territories in this study, Shandong (SD; 12.6 thousand), Jiangsu (JS; 10.8 thousand), and Hubei (HUB; 10.1 thousand) were the top three most populous provinces (Fig. 2A). The participants' median age was 32 ± 5.52 years (mean ± standard deviation), and the median gestational age was 13 weeks (range, 11 to 20 weeks). Within the NIPT cohort, 1,524 different bacteria (1,524/1,734), 170 eukaryotes (170/326), and 139 viruses (139/1,866) were identified out of the known species in the HMREFG. Among the 107,763 samples, only 2 samples contained no reads connected to any known microorganism by mapping the high-quality reads to HMREFG. Among the exogenous molecules detected in the cell-free DNA, more than 95% of DNA was from bacteria, 3% from eukaryotes, 0.4% from viruses, and the rest from unsorted microorganisms, mainly due to the cumulatively large genome sizes of bacteria and the resulting greater abundances of their DNA fragments. Among the

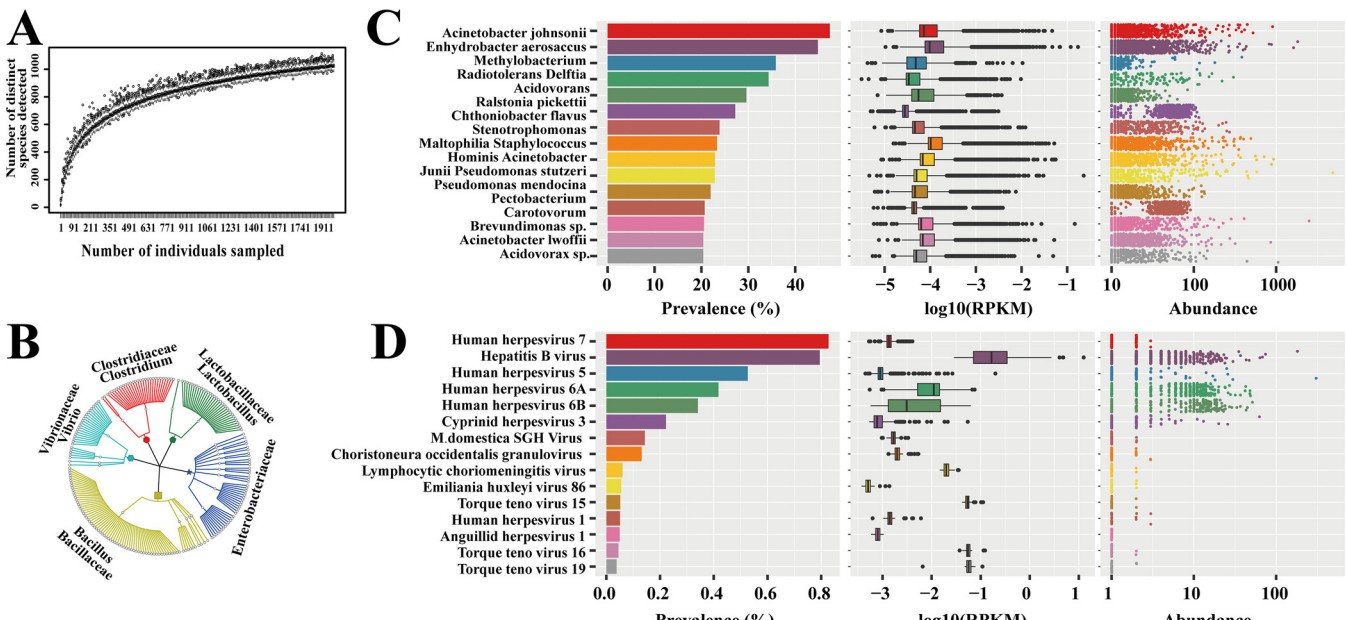

**FIG 1** Outline of peripheral blood microbiome analysis in this nationwide study. (A) Species counts with different sample sizes to check for potential saturation effect. (B) Top 5 families of detectable microorganisms from HMREFG. (C) Prevalence, log₁₀(RPKM), and absolute abundance values of the top 15 most frequently detected bacteria. (D) Prevalence, log₁₀(RPKM), and absolute abundance values of the top 15 most frequently detected viruses.

listed detectable microorganisms from HMREFG, *Bacillaceae* was the largest family, with *Bacillus* as the largest genus, followed by *Enterobacteriaceae*, *Lactobacillaceae*, *Clostridiaceae*, and *Vibrionaceae* (Fig. 1B).

In Fig. 1C and D, the top 15 bacteria and top 15 viruses are illustrated, together with their corresponding absolute abundances (read count) and relative abundances in log₁₀ reads per kilobase per million mapped reads (RPKM). Figure 1C lists the most frequently detected bacteria. Among 1,524 different bacteria identified in the study samples, *Acinetobacter johnsonii* was found in 47.4% of samples, *Enhydrobacter aerosaccus* in 35.9%, and *Delftia acidovorans* in 34.3%. Analysis of sequence diversity identified the presence of bacterial strains of *Acinetobacter* and *Pseudomonas*. Among 139 different viruses identified in the study materials, the two most frequent opportunistic viruses were human herpesvirus 7 (HHV-7) (0.85%) and hepatitis B virus (HBV) (0.8%),

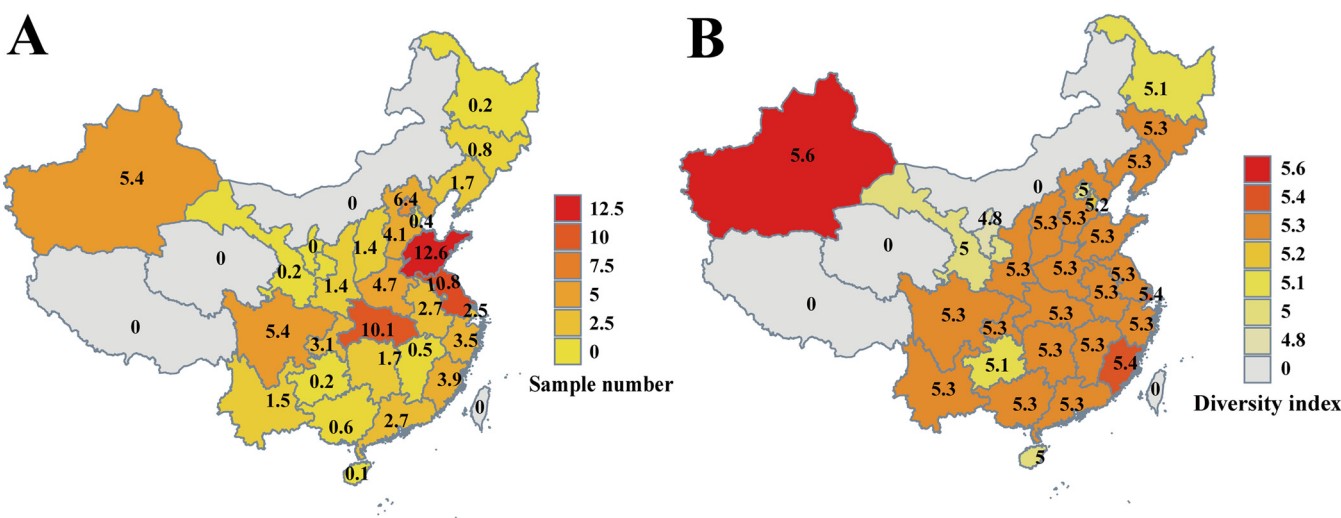

**FIG 2** The numbers of samples and differences in compositions of microbiota from peripheral plasma samples among different provinces in China. (A and B) Sample numbers (in thousands) (A) and Shannon diversity (B) at the province level across China.

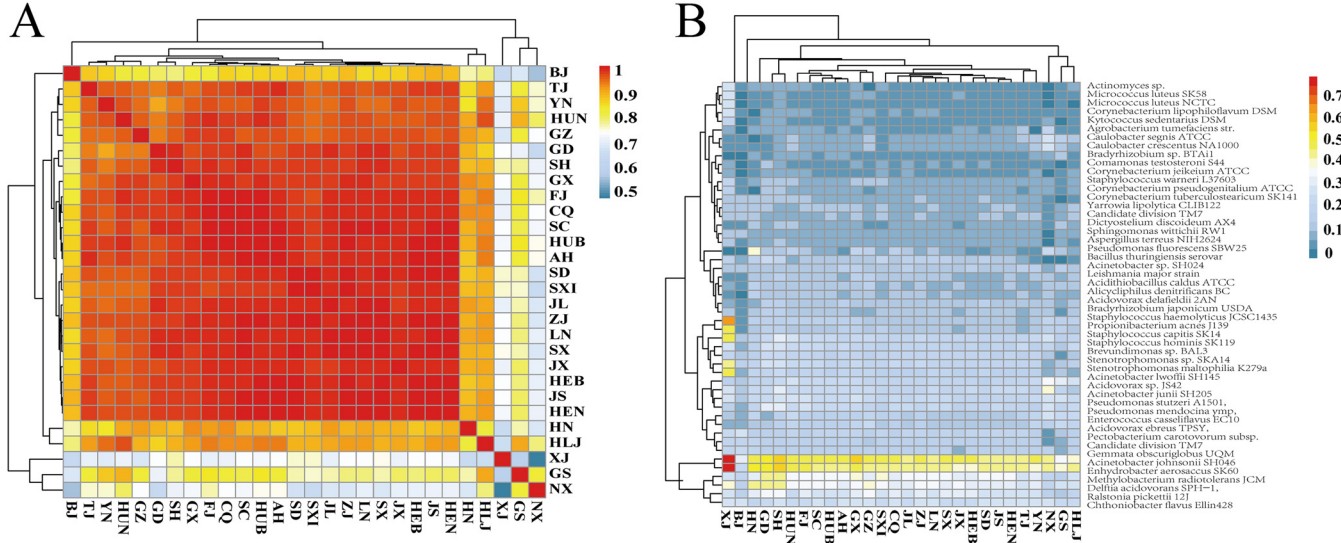

**FIG 3** Identification of top 50 bacteria detected as analyzed by NIPT at the province level in China. (A) Bacterial abundance similarity metric across provinces. (B) Abundance heatmap of the top 50 bacteria between provinces in China. *B. thuringiensis* serovar israelensis is listed as *B. thuringiensis* serovar. SX, Shanxi; XJ, Xinjiang; HEB, Hebei; BJ, Beijing; TJ, Tianjin; CQ, Chongqing; SC, Sichuan; GZ, Guizhou; HUB, Hubei; HUN, Hunan; YN, Yunnan; GX, Guangxi; GD, Guangdong; FJ, Fujian; JX, Jiangxi; SD, Shandong; AH, Anhui; HEN, Henan; LN, Liaoning; JL, Jilin; HLJ, Heilongjing; ZJ, Zhejiang; SH, Shanghai; JS, Jiangsu; HN, Hainan; NX, Ningxia; GS, Gansu.

listed in Fig. 1D. Analysis of sequence diversity identified the presence of both HHV-6 subtypes, A and B. The estimated proportions were 57% for subtype A and 43% for subtype B. HHV-6A and HHV-6B were identified in 0.4% and 0.3% of samples, respectively. A small proportion of individuals were found to carry sequences of other human herpesviruses, including herpes simplex virus 1 (HSV1) and cytomegalovirus (CMV; also known as HHV-5). Although HHV-8 was not on this top 15 list shown in Fig. 1D, it was also observed with low abundances.

**Geographic-area-associated genera of bacteria in plasma.** Among all the province-level territories in China, Shandong, Jiangsu, and Hubei were the top three contributors of cohort samples. The numbers of samples from other provinces are also included in the results shown in Fig. 2A. As shown by the results in Fig. 2B, we demonstrated noticeable differences in the compositions of microbiota amino acid residues in peripheral plasma samples among different provinces by Shannon index. The Xinjiang Uygur Autonomous Region showed the highest level of diversity in terms of microorganism species detected considering its sample size, whereas the capital city, Beijing (BJ), exhibited a relatively lower level of complexity in its microbiota profile. Whereas territories with fewer samples also showed reduced diversity estimates, this was indicated also by examining the saturation simulation presented in Fig. 1A, in which the number of detected species gradually stabilized when reaching 1,500 samples. In our case, territories like Ningxia (NX) and Gansu (GS) had relatively fewer samples included and, therefore, also showed lower diversity indexes.

The top 50 bacteria detected among the collected samples according to the levels of abundance are listed in Fig. 3B, and the regional similarity metrics are illustrated in Fig. 3A according to units of province, autonomous region, and municipality. After territories such as Ningxia (NX) and Gansu (GS) with low sample sizes were excluded, Beijing (BJ), Hainan (HN), and Xinjiang (XJ) showed rather distinct patterns. The heat map indicated that XJ was comparatively the least similar to all other territories, while Beijing was more similar to northern territories, such as Shandong (SD) and Tianjin (TJ), and less similar to southern territories, including Shanghai (SH), Fujian (FJ), and Guangdong (GD). Bacteria (including increased levels of *Propionibacterium acne*, *A. johnsonii*, and *E. aerosaccus*) distinguishing XJ from other territories could be identified in the results shown in Fig. 3B, whereas in Beijing, relatively low levels of bacteria most commonly detected in other territories were found. For Hainan, a clear increase in

*Bacillus thuringiensis* serovar israelensis could be seen, which might be related to its use as a biopesticide for its mosquitocidal activity, especially in tropical and subtropical areas, to prevent the public health problem caused by Dengue fever (11).

**TORCH panel analysis by NIPT across the country.** TORCH (*Toxoplasma gondii*, other, rubella virus, cytomegalovirus, and herpes simplex virus) denotes five congenital infections that are important targets for prenatal screening at 12 to 26 weeks of gestation. In this study, we screened the common infections associated with congenital anomalies based on the TORCH panel via NIPT sequencing, including target DNA segments from *T. gondii*, "other" organisms HBV, human parvovirus (HPV), and *Treponema pallidum* (causing syphilis), and CMV and HSV. *T. gondii* is not part of the HMP, and thus, no orgName or orgID is provided in Table 1. Rubella virus was not included as it is an RNA virus. Also, no syphilis-positive sample was detected in our study. The prevalences and abundances of other TORCH-related organisms are listed in Table 1. Additional data from other publications were also recorded to serve as a reference, in which the clinical frequency (ClinicalFreq) is predefined elsewhere, mostly as the rate of positive serological screening tests among the individual study populations, while population frequency (PopFreq) is the frequency of nonzero samples of a particular organism directly derived from the indicator matrix of the study population. While the exact level of correlation between our data and the previous studies is not directly accessible, it is obvious that HBV kept high prevalence in Chinese population.

Ningxia (NX), Hainan (HN), and Guangzhou (GZ) showed rather distinct patterns compared to other territories, which can be seen by the results in Fig. 4A. Geographically, the prevalence of TORCH organisms was apparently high in some territories (Fig. 4B to E). XJ and some other territories in Southwest China had the highest prevalences of CMV and *T. gondii*, which might be related to factors like low urbanization levels, agriculture-dominated economic structures, and the natural environment. The prevalence rates of HPV and HSV were slightly elevated in XJ but showed no evident change in other territories. The prevalences of Epstein-Barr virus (EBV) were elevated in Northeastern China but also showed no obvious change in other territories (Fig. 4F). The prevalence of HBV was higher in Southern China, especially in Hainan (Fig. 4G).

**Virus existence-dependent microorganism interactions.** The interactions between TORCH organisms and other cataloged bacteria are shown in Fig. 5. Among the HPV-positive samples, the microecology shifted to being significantly enriched with *Weissella paramesenteroides* and *Propionibacterium acne* (Fig. 5C). In the samples with detectable HSV amino acid residues, the microbiota showed increased enrichment (fold change of 7.44; $P < 0.001$) of *Bifidobacterium bifidum* (Fig. 5D). We also observed a similar increase (fold change of 6.96; $P < 0.001$) in the abundance of *P. acne*. Within the positive samples carrying *Toxoplasma gondii*, increased levels of *Pectobacterium carotovorum* and *Chthoniobacter flavus* were also observed (Fig. 5A). Their increased coexistence in *T. gondii*-positive samples suggests a complex human environment interaction. Within the positive samples carrying EBV, increased levels of HHV-6B and *P. carotovorum* were observed (Fig. 5D and E). No significant changes in other detected microbiota were noted when HBV or CMV was positive (Fig. 5B and F).

**Weighted coexistence network analysis of microbial communities.** Figure 6 shows a virus-bacteria interaction network. The node color signifies the module membership, and the node shape indicates the type of organism, with octagons representing bacteria and diamonds for viruses. The network is undirected with at least one virus-type node. The name-to-identification number (ID no.) mapping is shown in Table S1 in the supplemental material.

The largest green cluster is composed of many different genera of bacteria (12), with one of the most widely studied and medically important bacteria, *Streptomyces* (ID no. 246), and Streptomyces phage (ID no. 247) as its center. The coexistence relationship is therefore largely related to the soil-based bacteria and the ecosystem of other microorganisms in which *Streptomyces* is known to play a critical role (13).

The pink nodes clustered within the beige module hold the largest number of viruses, mainly bacteriophages. A smaller isolated cluster of bacteria is solely connected to *Haemophilus*

**TABLE 1** Population-level statistics regarding the TORCH panel[a]

| Official name | HMP orgID | HMP orgName | ClinicalFreq (%) | RPKM value | | PopFreq |
| --- | --- | --- | --- | --- | --- | --- |
| | | | | Reference | Maximum | |
| *Toxoplasma gondii* | NA | NA | 2.2[b] | 0 | 0.000116 | 0.00409856 |
| Rubella virus | VIRL\|gi\|9790308\|ref\|NC_001545.1\| | Rubella virus, complete genome | 1.6[b] | NA | NA | NA |
| Cytomegalovirus | VIRL\|gi\|155573622\|ref\|NC_006273.2\| | Human herpesvirus 5, complete genome | 1.6[b] | 0 | 0.199486 | 0.00526411 |
| Herpes simplex virus | VIRL\|gi\|9629378\|ref\|NC_001806.1\| | Human herpesvirus 1, complete genome | 1.6[b] | 0 | 0.00606 | 0.00050635 |
| | VIRL\|gi\|9629267\|ref\|NC_001798.1\| | Human herpesvirus 2, complete genome | 1.1[b] | 0 | 0.005336 | 5.73225E−05 |
| Human parvovirus B19 | VIRL\|gi\|9632996\|ref\|NC_000883.1\| | Human parvovirus B19, complete genome | 8.1[b] | 0 | 1.904969 | 3.8215E−05 |
| *Treponema pallidum* | BACT_1204 | *Treponema pallidum* subsp. *pallidum* SS14 | 4.36[c] | 0 | 0 | 0 |
| Hepatitis B virus | VIRL\|gi\|21326584\|ref\|NC_003977.1\| | Hepatitis B virus, complete genome | 35.614[c] | 0 | 12.60268 | 0.00794872 |

[a]The TORCH organisms include *Toxoplasma gondii*, others (*Treponema pallidum* [causing syphilis], hepatitis B virus [HBV], and human parvovirus B19), rubella virus, cytomegalovirus (CMV), and herpes simplex virus (HSV). *T. gondii* is not part of the Human Microbiome Project (HMP), and thus, no orgName or orgID is provided. Rubella virus was not detected because, as an RNA virus, it does not fit in the NIPT library protocol. Clinical frequency (ClinicalFreq) is predefined elsewhere, mostly as the rate of positive serological screening tests among the individual study populations, while population frequency (PopFreq) is the frequency of nonzero samples of a particular organism directly derived from the indicator matrix of the study population. NA, not available.

[b]Data are taken from reference 16.

[c]Data are taken from reference 17.

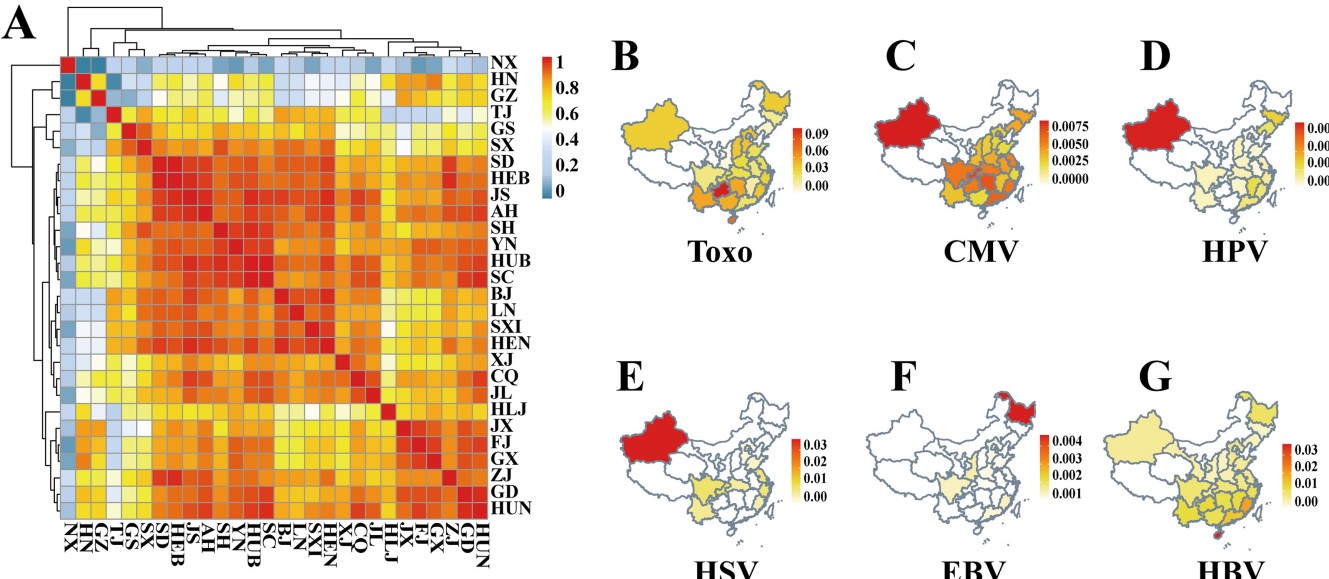

**FIG 4** Heat map and geographic maps of organism prevalences from TORCH analysis by NIPT at the province level in China. (A) Virus abundance similarity metric across provinces. Province abbreviations are listed in the legend to Fig. 3. (B to G) Prevalences of *Toxoplasma* (Toxo)-related (B), CMV-related (C), HPV-related (D), HSV-related (E), EBV-related (F), and HBV-related (G) microorganisms across provinces in China.

phage (ID no. 117), including *Aggregatibacter*, *Neisseria*, *Pasteurella*, and *Veillonella*. There has been accumulating evidence showing shared patterns and conditions in the colonization and/or pathogenesis of these bacterial species (14), and *Haemophilus* phage could be a common bacterial inhibition agent.

Vibrio phage (ID no. 272), originally belonging to the green cluster which centered on Streptomyces phage (ID no. 247), appears to be a connecting hub to another smaller beige cluster with *Serratia* (ID no. 225), *Vibrio* (ID no. 271), and *Gardnerella* (ID no. 107). The beige cluster is mainly related to a group of bacteriophages that includes Enterobacteria phage (ID no. 94), Kluyvera phage (ID no. 129), Morganella phage (ID no. 156), porcine endogenous retrovirus E (ID no. 191), *Salmonella* phage (ID no. 221), and Sodalis phage (ID no. 233). Also connecting to the hub, *Leptolyngbya* (ID no. 139) has recently been found to exhibit an inhibitory effect on *Vibrio* (ID no. 271) (15), and therefore, it might share a similar cohabitation pattern with Vibrio phage (ID no. 272). The cyanobacteria and the waterborne nature of many bacteria suggest seafood as their most likely origin, but not to the exclusion of factors like frequent environmental contact of the host.

Actinomyces phage (ID no. 11), which coexists with another cluster of bacteria and 2 other bacteriophages, Propionibacterium phage (ID no. 195) and *Staphylococcus* phage (ID no. 240), is the center of the light blue module. The cluster contains a group of anaerobic and microaerophilic bacteria, such as *Propionibacterium* (ID no. 194), *Streptobacillus* (ID no. 244), *Staphylococcus* (ID no. 239), and *Anaerococcus* (ID no. 26), suggesting the gastrointestinal tract and oropharynx as their origin.

## DISCUSSION

NIPT for fetal aneuploidy by scanning cfDNA in maternal plasma samples has become a popular prenatal genetic test in China. In addition, cfDNA can also be used to characterize the signatures of the peripheral blood microbiome and analyze the burdens of bacteria, viruses, and other microbes quantitatively. Population-based studies on the microbiome community may provide a microbial baseline to facilitate pathogen identification in clinical practice. The distribution of viral DNA in blood plasma samples has been detected via NIPT at the population level (10), which brings a novel insight into viral prevalences and potential public health burdens. Given our low-depth sequencing pipeline, which might have missed organisms of lower abundance in the

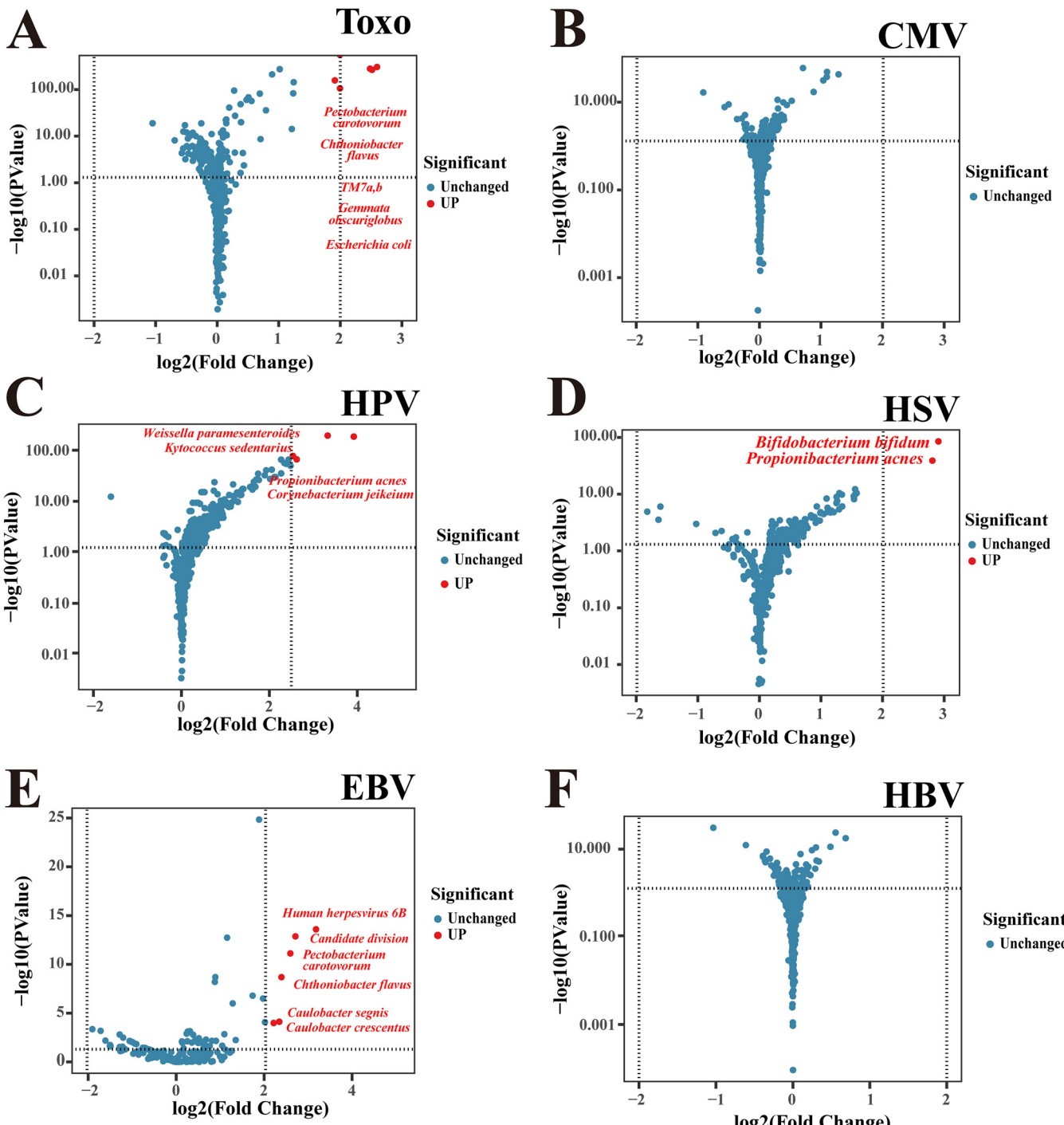

**FIG 5** Volcano plots show abundances of cataloged bacteria between samples with and without TORCH organisms. (A to F) Volcano plots of differential abundance analysis between samples with and without *Toxoplasma* (Toxo) (A), CMV (B), HPV (C), HSV (D), EBV (E), and HBV (F).

samples, the positive rate (1.1%) of HSV was slightly higher than those previously reported in China; the positive rate of HBV was 0.8% in our current study, much lower than those reported in China (27%) and worldwide (35.6%) (16–18), which may be explained by the wide hepatitis B vaccine coverage in China and the good health status of our subjects (healthy women aged 20 years to mid-30s and living in urban areas). HBV DNA in plasma samples had a higher prevalence in Southern China than in Central and Northern China, which agreed with a previous study (10). Otherwise, we observed an increased enrichment of *B. bifidum* in the HSV-positive population; *B.*

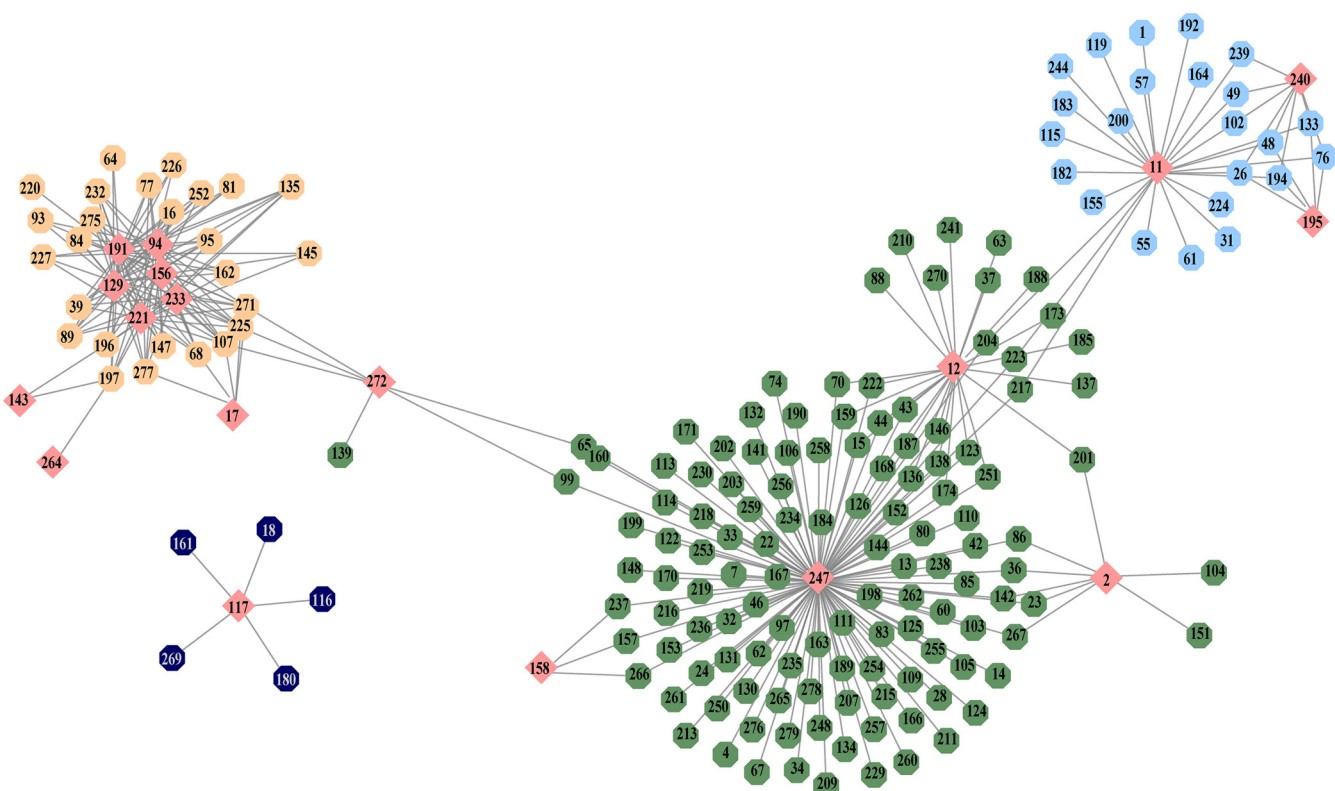

**FIG 6** Weighted virus-bacterium coexistence network. The node color signifies the module membership. The node shape indicates the type of organism, with octagons representing bacteria and diamonds for viruses. The name-to-identification number mapping is shown in Table S1.

*bifidum* is considered probiotic and is frequently used to help maintain and restore the normal balance of the gut microbiome (19). These findings further suggested the possible gut origin of microbial cfDNA fragments in plasma samples. In our current study, we performed weighted coexistence network analysis of microbial communities to identify coexistence of viruses and bacteria in the population, and such coexistence revealed the complexity of the microbiome in peripheral blood and should be considered a key factor in the interpretation of microbial cfDNA sequencing results.

In this study, a quantitative baseline of microbial DNA in peripheral blood was established via NIPT in pregnant women. The resulting range of detected exogenous DNA could be considered a normal reference data set. Thus, infectious cases could be identified when a certain microorganism is overly enriched. On the other hand, we also hypothesize that, without a clinically defined infection, all samples detected with microbial DNA do not necessarily indicate that the microbe has caused damage or been harmful to the host, and thus, we may use the indicator matrix as the input to calculate population-level frequencies for any detected microorganism. Collectively, the data provide the possibility of identifying microbiota and metabolic and regulatory networks, as well as correlations between microbial community structure and health and special conditions.

This study also showed large geographic diversities of the microbiota. Territories in China at similar latitudes showed the same cluster distribution features. The prevalence rates of major viruses, including HBV, CMV, EBV, human parvovirus (HPV), and HSV, also differed among different areas. Interestingly, two blood viromes from Europe and China showed different viral distributions, in which the European virome appeared to have a different viral distribution than was found in the pregnant Chinese women analyzed in a previous study (10, 20). Although this study used the neglectful data from NIPT, which was not optimized for cfDNA microbiome reference study, the potential pollution from skin puncture and vacuum blood collection tubes was not evaluated,

and direct comparisons were difficult due to the variations in sequencing protocols, our current findings showed certain consistencies with the previous study in China (10, 21). HBV was significantly enriched in both our current study and that of BGI; nevertheless, human parvovirus B19 DNA showed a lower prevalence in our study than in that of BGI (10).

Accordingly, to characterize the ecology of human-associated microbial communities, this population-based study analyzed blood microbial information in this largest-scale set of distinct, clinically relevant body habitats to date. We found the diversity and abundance of each habitat's signature microbes. These results delineated the normal range of structural and functional configurations in the microbial communities of a healthy population, which may enable future characterization of the epidemiology, ecology, and translational applications of the human microbiota. Furthermore, these results suggested that different countries and regions may have distinct normal reference intervals of microbes in peripheral blood, which may be influenced by diet, geography, and environmental factors. Therefore, our current study timely established a reference range of characteristic peripheral blood microorganisms. Application of the data maybe helpful for early detection and interpretation of infectious diseases, especially regional epidemic diseases, and provide pathogenic evidence for clinical diagnosis, which needs to be investigated and proved by a series of research studies.

**Conclusion.** Human microbiota may present in the peripheral blood in the form of microbial cfDNA residues, with notable diversity and coexistence. Therefore, cfDNA-based blood microbiota may be reflective of human microbiota, including gut microbiota, and also serves as a proxy of immune response in the presence of chronic or acute exogenous pathogenic stimulants. With our microbial cfDNA baseline data set constructed with peripheral blood samples from a phenotypically healthy population, a notably elevated level of microbial amino acid residues or a perturbed cohabitation pattern of a collection of microorganisms in blood might be indicative of a relevant disease status. Such perturbances, after being quantified, may be adapted as potential biomarkers and exert diagnostic value in the future.

## MATERIALS AND METHODS

**Ethical approval.** The study was approved by the Ethics Committee of Beijing Hospital (2020BJYYEC-046-01), and written informed consent was provided by all participants.

**Study design.** A nationwide retrospective meta-analysis was performed based on data collected from 3 January 2016 to 31 December 2017 by continuously aggregating NIPT data of pregnant women undergoing testing. In total, we included data from 107,763 samples in this study. All patients gave consent to the collection of blood samples and the subsequent clinical genetic testing and to its anonymized usage for academic research. This study was reviewed and approved by the Institutional Review Board of Beijing Hospital, China.

**DNA extraction and sequencing.** Peripheral venous blood (5 mL) from each patient was preserved and delivered to the laboratory in EDTA tubes (Sekisui, Tokyo, Japan) or Streck tubes (La Vista, NE, US). Plasma was separated after 2 rounds of centrifugation (22) and stored at $-80°C$ until DNA extraction. Cell-free DNA was extracted from plasma according to standard commercial protocols described in previous publications (22–24). The sequencing library was prepared using an NMPA certified kit (registration no. 20173400331), and subsequently, 4.2 million single-end reads of 40 bp were generated for each sample library using NextSeq 550AR (Annoroad Gene Tech, Beijing, China). All procedures were performed in a standard negative-pressure laboratory with constant temperature and humidity.

**Preprocessing methods.** Sequencing data from standard NIPT screening were collected and reanalyzed using the following pipeline. For each sample library, reads with perfect matches of the first 35 bp to the human genome (hg19) were initially excluded while being used for routine NIPT screening with an in-house aligner. The remaining reads with mismatches to hg19 were then identified in the second alignment run using Burroughs-Wheeler Aligner (BWA) after adapter trimming. At the end of the first two alignment runs, we considered the remaining reads to be a collection of nucleic acid molecules of exogenous origin.

We downloaded the human microbiome reference genome (HMREFG) assembled by the NIH Human Microbiome Project (HMP; http://hmpdacc.org/HMREFG/). The remaining reads were aligned to those genome-level HMREFG reference sequences, with perfect matches only; after removal of PCR duplicates, the uniquely mapped reads were counted to derive an abundance matrix for each sample library. In order to remove potential contaminants during the experiment, and also as a standard internal quality control (QC) step for aerosol monitoring, we set up a blank sample in every batch run. In this pipeline, we trimmed the species, if any, present in the negative QC sample from other real samples of the same batch. All these samples passed the internal quality control successfully without other contaminants.

**Abundance matrix.** Based on all the sample libraries collected, an abundance matrix was created with rows and columns representing microorganisms and sample IDs, respectively. The raw count matrix was then converted into an RPKM (reads per kilobase per million mapped reads)-like normalized abundance matrix by

dividing the genome length of each organism while taking into account the sequencing depth per sample. A logical indicator matrix can be created on the go if for a particular library and a named organism the normalized abundance is greater than zero. A taxonomic tree of organisms from the top 5 most abundant families was illustrated using GraPhlAn (25). The normalized abundance matrix was then aggregated into a regional frequency matrix by summing up the sample columns of the region from the indicator matrix.

**Detection model parameter estimation.** With the large set of normal low-depth whole-genome-sequencing (WGS) data from plasma samples, we devised a detection model similar to that described in Grumaz et al. (1), where it is assumed that the abundance of any individual microorganism detected in plasma follows a Poisson distribution, for which the single mean/variance parameter lambda can be estimated from a set of normally unaffected samples sequenced at a given level. With the estimated model, for each organism, one can predict the likelihood of any test sample that falls out of the normal range, which could be an indicator of sepsis.

However, for WGS-based NIPT, any individual library is sequenced at a rather shallow level, typically about $0.1\times$ coverage of the human genome. It is not straightforward to derive such a model estimate based on a single insufficiently sequenced library. In our current study, we used population-wide data to simulate WGS data, which were sequenced at a much higher level. Individual libraries were sampled and merged with different sizes (from 10 to 1000), corresponding to roughly $1\times$ to $100\times$ coverage of the human genome. For each size level, 10,000-permutation resampling was performed to generate 10,000 simulated libraries. Poisson model parameters were then estimated using the 10,000 artificial samples for all microorganisms listed in this study. Then, 1,000 such permutation runs were similarly performed to assess the variation of the estimates.

**Statistical tests for the detection model.** We used the trained prediction model parameter to evaluate the probability of infection (above reference-level abundance) for each validation subject. Two statistics, including the sepsis-indicating quantifier (SIQ), adopted from the Grumaz et al. study (1), and the $P$ value obtained with the one-sided Poisson test, were generated. The lambda parameter ($\tilde{\lambda}$) used in both models was estimated from permutation output using the data from the reference panel with the following equation:

$$\tilde{\lambda}_i = 1.644854 \times \text{SD}(\lambda_i) + \text{median}(\lambda_i)$$

where $i$ indicates the sequencing depth groups formed by permutation and aggregation and SD and median are functions summarizing the standard deviation and median of all permutation sample estimates of the lambda parameter at a given depth. The final parameter estimate $\tilde{\lambda}$ is the 95% upper confidence interval of the permutation estimates.

**Simulation of potential saturation effect.** We also investigated the possible saturation effect when sampling from a large collection of homogenously generated sequencing data with our certified experimental protocol, so as to rule out extensive contamination in commercial reagents or common environmental factors. We randomly drew and combined $N$ (from 1 to 2,000) individual samples to form 1,000 pseudosamples for each $N$; also, the number of distinct species detected among these pseudosamples was counted, and the standard deviations of the species count were calculated over 1,000 permutation runs for each sample size $N$. Such a test could indicate a potentially saturated enrichment of most organisms detected due to a common source of DNA contamination, when early equilibrium was reached for the species count irrespective of the increasing combined sample size.

**Geographic distribution of bacteria and viruses.** We analyzed the abundances and complexity of different microorganisms in each province. The Shannon index based on microbial species profile was calculated for each province. Then, we explored the geographic distribution of viruses detected in the studied populations throughout mainland China by mapping the prevalences of viral sequences among 28 province-level territories.

**Differential abundance analysis.** We analyzed the differences in the abundance distributions of the microbiome communities given whether certain species of interest were positively detected to see if there were conditionally dependent interactions between some well-known species and others in HMREFG. For any given organism of interest, a group of positive samples were selected by checking the corresponding indicator matrix row; in addition, a negative set of samples were defined as samples that did not contain any read of all these species of interest. All other species in the raw abundance matrix other than these organisms of interest were tested between the two sets of samples using DESeq2 (26). Top-ranking interacting species were illustrated in the volcano plots for each species of interests. An empirical cutoff of a fold change greater than 2 and a $P$ value smaller than 1E−4 was used.

**Weighted microorganism coexistence network analysis.** With the collected samples, a weighted microorganism coexistence network was constructed manually using the WGCNA package in R (27). To simplify the analysis, for each genus, the species with the highest sum of RPKM values was selected to be the representative of that genus. The normalized RPKM value of each target organism per library was used as input. All data were first hierarchically clustered using the average linkage function and common Euclidean distance. Organisms with invariant abundance profiles across samples were excluded from the downstream analysis. WGCNA methodology, rather than using traditional distance or correlation-based similarity measures, utilizes the topological overlap matrix (TOM) $\Omega = [\omega_{ij}]$,

$$\omega_{ij} = \frac{a_{ij} + \sum_u a_{iu} a_{uj}}{\min\left\{\sum_u a_{iu}, \sum_u a_{ju}\right\} + 1 - a_{ij}} , \quad a_{ij} = |cor(x_i, x_j)|^\beta$$

which not only estimates the closeness between a pair of nodes but also reflects each nodes' relative interconnectedness to all other nodes. Thus, a TOM-based distance matrix may be a more robust and powerful measurement in building such a network. It was then necessary to select an appropriate value for the soft

thresholding power $\beta$ (28). After the scale-free topology model fit with candidate $\beta$ values for each set of expression profiles was manually inspected, the minimal $\beta$ value giving a coefficient of determination ($R^2$) higher than 90% was adopted. In the next step, modules of microorganisms were formed by hierarchical clustering with the aforementioned TOM distance matrix, with an empirically specified minimal module size. Modules were further merged based on the dissimilarity between their co-existence microorganisms, which was defined as the first principal component of all targeted organisms within each module. Pairs of modules with dissimilarity below 0.2, recommended by the WGCNA author, were then merged. Such a threshold in turn corresponds to a correlation of 0.8 between module eigenorganisms. To further track down the key player in each module and the regulation network, module membership (MM), defined as the correlation of the abundance profile and each module eigenorganism, was applied.

The resulting TOM-based network was exported for visualization with Cytoscape (29). With nodes representing microorganisms identified, an edge was defined as the TOM-based similarity between any pair of two organisms. Different shapes representing the nodes were used to distinguish bacteria and viruses.

Unless otherwise stated, all aforementioned statistical analysis was carried out in the statistical software R (30).

**Data availability.** All sequencing data are managed by Annoroad Gene Technology Co., Ltd., and are available upon reasonable request according to Chinese regulatory rules. Direct sharing of individual raw genetic data from this project is not allowable according to the Human Genetic Resources Administration of China (HGRAC). However, we have extracted summary data only related to microbiota and made such anonymized summary data available to the research community. The partially processed input data for the analysis have been submitted to Open Archive for Miscellaneous Data (https://ngdc.cncb.ac.cn/omix/release/OMIX001022), the access to which might be subject to individual verification. Also, to support reproducible research, the analytical scripts related to this study have been made available via GitHub (https://github.com/yd23/nipt-microbial-codes).

## SUPPLEMENTAL MATERIAL

Supplemental material is available online only.
**SUPPLEMENTAL FILE 1**, XLSX file, 1.8 MB.

## ACKNOWLEDGMENTS

This work was supported by the National Key Research and Development Program of China (grants no. 2018YFC2000505 and 2020YFC2009003), CAMS Innovation Fund for Medical Sciences (grant no. 2021-I2M-1-050), the Applied Research Program of Capital Clinical Features (grant no. Z18110001718172), National Natural Science Foundation of China (grant no. 81870013), and Natural Science Foundation of Beijing (grant no. 7202178). We appreciate all the financial support from the aforementioned funding agencies.

Special thanks to the staffs of all clinical laboratories involved and their excellent experimental work to produce the sequencing data.

We declare that we have no competing interests.

Yanming Li, Fei Xiao, Xunliang Tong, Xiaowei Yu, and Yang Du contributed to the conception and design of the study and interpretation of the results and drafted the manuscript. Xunliang Tong, Yang Du, Fei Su, and Yunshan Liu contributed to the acquisition of the data and revision of the manuscript for important intellectual content. Ye Liu, Hexin Li, Yunshan Liu, Kai Mu, Qingsong Liu, Hui Li, Jiansheng Zhu, and Hongtao Xu performed the statistical analysis and revised the manuscript for important intellectual content. All authors read and approved the final manuscript.

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
