## [Reviewer comments · Microbiology Spectrum]

Microbiology Spectrum

The peripheral blood microbiome analysis via NIPT reveals the complexity of circulating microbial cell-free DNA baselines

Xunliang Tong, Xiaowei Yu, Yang Du, Fei Su, Ye Liu, Hexin Li, Yunshan Liu, Kai Mu, Qingsong Liu, Hui Li, Jiansheng Zhu, Hongtao Xu, Fei Xiao, and Yanming Li

Corresponding Author(s): Fei Xiao, Beijing Hospital

Review Timeline:

Submission Date:	February 2, 2022
Editorial Decision:	March 7, 2022
Revision Received:	April 16, 2022
Accepted:	May 2, 2022

Editor: Wei-Hua Chen

Reviewer(s): The reviewers have opted to remain anonymous.

Transaction Report:

DOI: <https://doi.org/10.1128/spectrum.00414-22>

March 7, 2022

Dr. Fei Xiao
Beijing Hospital
Beijing
China

Re: Spectrum00414-22 (The peripheral blood microbiome analysis via NIPT reveals the complexity of circulating microbial cell-free DNA baselines)

Dear Dr. Fei Xiao:

Thank you for submitting your manuscript to Microbiology Spectrum. Your manuscript has been seen by two external experts. As you will see below, their comments are in general positive. However, they, especially reviewer #1, also raised substantial concerns; in particular, the reviewer found conclusions that are not supported by the results. In addition, the manuscript can be improved by English language editing service, as one reviewer pointed out.

Thus, I would like to invite you to submit a revised version of your manuscript, which will be submitted to another round of review, if all the reviewer's concerns are sufficiently addressed.

Link Not Available

Sincerely,

Wei-Hua Chen

Journals Department
Reviewer comments:

Reviewer #1 (Comments for the Author):

In the manuscript "The peripheral blood microbiome analysis via NIPT reveals the complexity of circulating microbial cell-free DNA baselines", Tong et al define microbial cell-free DNA (cfDNA) in >100K healthy individuals to provide a reference for

interpreting levels in disease. Cell-free DNA offers the potential to diagnose infections when direct sampling is invasive or not possible. Microbial sequences were detected in nearly every sample and were predominantly bacterial in origin. The authors defined the microorganisms and potential interactions between bacterial and viral communities were defined. Significant regional diversity in prevalence of viruses were identified. They also constructed a weighted micro-organism co-existence network.

The methods are clearly stated, the figures are clear, and the figure legends are appropriate. The microecology shifts in bacterial sequence abundance detected by the presence of TORCH virus genomes is particularly interesting. One of the study goals stated in the abstract is to establish the microbial cfDNA baseline in healthy individuals. In the methods they describe establishing a method to calculate the likelihood of any test sample that falls out of the normal range, using NIPT WGS data which is sequenced at a lower depth, for the future use of diagnosing sepsis. In the Availability of data and materials section, the authors state that all sequencing data that is held by a company will be shared upon request. Adding that their analysis code will also be available to shared would allow the analysis that is visualized in the figures in this manuscript to serve as a true reference dataset for investigators studying samples from diseased patients.

The conclusions are not fully supported by the data presented and should be reworded. The conclusion that microbial cfDNA is a potential biomarker of disease is accurate. Referring to the presence of cfDNA in the blood as blood microbiota risks creating confusion, as the presence of DNA does not necessarily imply the presence of intact bacterial microbiota. "Blood microbiota may represent or contribute to the first step in the kinetics of disease" is speculative and not a conclusion of the study. Considering this study is in healthy individuals, the presence of gut commensals doesn't indicate pathogens driving disease, and the presence of viral sequence reflects chronic infections rather than the first step of acute infection.

Minor comments:

- HHV8 is mentioned in line 237 in reference to figure 1D, but not listed in the figure.
- Suggest removing the subjective descriptor of "dramatic" when describing the regional differences in Shannon's diversity (line 241). The results show the majority are 5.3 and range from 5 to 5.6. Shannon diversity is somewhat contextual and dramatic in disease conditions is used to refer to differences between >5 and <1 .
- Line 263. This reviewer cannot find "Bacillus thuringiensis serovar" on Figure 3B that is stated as a key finding in figure.
- Line 277. Can ClinicalFreq be defined so readers can properly interpret ClinFreq=36% and PopFreq=0.008 for HBV?
- Line 294. If the authors consider the increase in the abundance of Propionibacterium acne to be "slight" compared with Bifidobacterium bifidum, this should be described further as the fold change and p values are similar on Fig. 5D.
- Fig. 5E. Authors should confirm that "candidate division" indicated on the figure is intended.
- Could the authors reword line 312 "Module pink holds the largest number of viruses, mainly bacteria phages". All viruses appear pink, so it's not clear what module refers to.
- Two blue clusters appear in Fig 6. Consider adding "light blue" to describe the cluster in line 329, and dark blue to line 313 describing the bacterial cluster centered on Haemophilus phage.

Reviewer #2 (Public repository details (Required)):

A set of sequencing data

Reviewer #2 (Comments for the Author):

Thank you for inviting me to review the manuscript 'The peripheral blood microbiome analysis via NIPT reveals the complexity of circulating microbial cell-free DNA baselines'.

My general impression is that the described study will be of interest for the research community. Innovative research data are described.

Major shortcomings are:

Title:NIPT abbreviation should be explained in the title.

Abstract:

The abstract is not fully clear. The aims of the study, methodology and research problems that the authors investigated are not very clear. The major findings or trends are only hinted at. Brief and more concrete description regarding approach and experimental design, materials, concrete results, conclusions and interpretation should be described.

Introduction:

Lines 59-63: It would be useful for the readers to understand the meaning of the term used by the authors ,baseline microbiome' and if they mean also ,core microbiome'.

Line 63-64: 'We also investigated the bacteria and virus ecologies in human blood as well as the co-existence and interactions between these microbes.' - Results demonstrate that the study was limited to the evaluation of the microbial abundance in serum samples and that some indirect (statistical) interactions could be supposed.

Materials:

The raw sequencing data are not publicly available. The developed and applied statistical algorithms are not available for open access.

Results:

Line 308-311: Please clarify the meaning or rephrase ,... medically utilized bacteria Streptomyces (246) and Streptomyces phage (247) as its center. Also which hypothesis you mean?

Figure 6. Specify the meaning of the numbers in the squares and octagons. A supplementary file would be useful.

All figures and table are appropriate.

Discussion:

Line 383-387: 'our current study timely established a reference range of characteristic peripheral blood microorganisms, which is helpful for early detection and interpretation of infectious diseases, especially regional epidemic diseases, and provides the pathogenic evidence for clinical diagnosis.' - This statement sounds logic but it is not supported by concrete data. It sounds like a declaration. Please rephrase.

I agree with the conclusion - 'Furthermore, these results suggested that different countries and regions may have distinct normal reference intervals of microbes in peripheral blood, which may be influenced by diet, geography, and environmental factors.' The study results give support.

Staff Comments:

Preparing Revision Guidelines

Please return the manuscript within 60 days; if you cannot complete the modification within this time period, please contact me. If you do not wish to modify the manuscript and prefer to submit it to another journal, please notify me of your decision immediately so that the manuscript may be formally withdrawn from consideration by Microbiology Spectrum.

Spectrum00414-22

The peripheral blood microbiome analysis via NIPT reveals the complexity of circulating microbial cell-free DNA baselines

Comments and Suggestions for the Author:

In the manuscript “The peripheral blood microbiome analysis via NIPT reveals the complexity of circulating microbial cell-free DNA baselines”, Tong et al define microbial cell-free DNA (cfDNA) in >100K healthy individuals to provide a reference for interpreting levels in disease. Cell-free DNA offers the potential to diagnose infections when direct sampling is invasive or not possible. Microbial sequences were detected in nearly every sample and were predominantly bacterial in origin. The authors defined the microorganisms and potential interactions between bacterial and viral communities were defined. Significant regional diversity in prevalence of viruses were identified. They also constructed a weighted micro-organism co-existence network.

The methods are clearly stated, the figures are clear, and the figure legends are appropriate. The microecology shifts in bacterial sequence abundance detected by the presence of TORCH virus genomes is particularly interesting. One of the study goals stated in the abstract is to establish the microbial cfDNA baseline in healthy individuals. In the methods they describe establishing a method to calculate the likelihood of any test sample that falls out of the normal range, using NIPT WGS data which is sequenced at a lower depth, for the future use of diagnosing sepsis. In the Availability of data and materials section, the authors state that all sequencing data that is held by a company will be shared upon request. Adding that their analysis code will also be available to shared would allow the analysis that is visualized in the figures in this manuscript to serve as a true reference dataset for investigators studying samples from diseased patients.

The conclusions are not fully supported by the data presented and should be reworded. The conclusion that microbial cfDNA is a potential biomarker of disease is accurate. Referring to the presence of cfDNA in the blood as blood microbiota risks creating confusion, as the presence of DNA does not necessarily imply the presence of intact bacterial microbiota. “Blood microbiota may represent or contribute to the first step in the kinetics of disease” is speculative and not a conclusion of the study. Considering this study is in healthy individuals, the presence of gut commensals doesn’t indicate pathogens driving disease, and the presence of viral sequence reflects chronic infections rather than the first step of acute infection.

Minor comments:

- HHV8 is mentioned in line 237 in reference to figure 1D, but not listed in the figure.
- Suggest removing the subjective descriptor of “dramatic” when describing the regional differences in Shannon’s diversity (line 241). The results show the majority are 5.3 and range from 5 to 5.6. Shannon diversity is somewhat contextual and dramatic in disease conditions is used to refer to differences between >5 and <1.
- Line 263. This reviewer cannot find “*Bacillus thuringiensis* serovar” on Figure 3B that is stated as a key finding in figure.
- Line 277. Can ClinicalFreq be defined so readers can properly interpret ClinFreq=36% and PopFreq=0.008 for HBV?
- Line 294. If the authors consider the increase in the abundance of *Propionibacterium acne* to be “slight” compared with *Bifidobacterium bifidum*, this should be described further as the fold change and p values are similar on Fig. 5D.

- Fig. 5E. Authors should confirm that “candidate division” indicated on the figure is intended.
- Could the authors reword line 312 “Module pink holds the largest number of viruses, mainly bacteria phages”. All viruses appear pink, so it’s not clear what module refers to.
- Two blue clusters appear in Fig 6. Consider adding “light blue” to describe the cluster in line 329, and dark blue to line 313 describing the bacterial cluster centered on *Haemophilus* phage.

April 10th, 2022

Editors

Microbiology Spectrum

Dear Editors:

Here within enclosed is our revised manuscript titled “A population-based metagenomics analysis of peripheral blood microbiome via NIPT in China”. (Spectrum00414-22) for your consideration in ***Microbiology Spectrum***.

We gratefully appreciate for the reviewers’ valuable suggestions. Each suggestion and comment brought forward by the reviewers was accurately incorporated and carefully considered, which helped us to improve the manuscript substantially. In the following pages we provide a point-to-point response to each comment from you and two reviewers.

Based on the reviewer’s suggestions, we added more information and the limitations on the interpretation of cell free DNA sequencing results. We also modified the statements that might cause ambiguity as required. We hope that the revised manuscript and our accompanying responses will be sufficient to make our manuscript suitable for publication in ***Microbiology Spectrum***.

Sincerely,

Fei Xiao, M.D. Ph.D.
Director, Clinical Biobank
Professor, Laboratory of Cell Biology
Beijing hospital
#1 Dahua Road, DongCheng District
Beijing, China 100730
Phone: 0086-10-58115083
E-mail: xiaofei3965@bjhmoh.cn

Responses to the 1st Reviewer's comments:

Major Comment 1: In the manuscript "The peripheral blood microbiome analysis via NIPT reveals the complexity of circulating microbial cell-free DNA baselines", Tong et al define microbial cell-free DNA (cfDNA) in >100K healthy individuals to provide a reference for interpreting levels in disease. Cell-free DNA offers the potential to diagnose infections when direct sampling is invasive or not possible. Microbial sequences were detected in nearly every sample and were predominantly bacterial in origin. The authors defined the microorganisms and potential interactions between bacterial and viral communities were defined. Significant regional diversity in prevalence of viruses were identified. They also constructed a weighted micro-organism co-existence network.

The methods are clearly stated, the figures are clear, and the figure legends are appropriate. The microecology shifts in bacterial sequence abundance detected by the presence of TORCH virus genomes is particularly interesting. One of the study goals stated in the abstract is to establish the microbial cfDNA baseline in healthy individuals. In the methods they describe establishing a method to calculate the likelihood of any test sample that falls out of the normal range, using NIPT WGS data which is sequenced at a lower depth, for the future use of diagnosing sepsis. In the Availability of data and materials section, the authors state that all sequencing data that is held by a company will be shared upon request. Adding that their analysis code will also be available to shared would allow the analysis that is visualized in the figures in this manuscript to serve as a true reference dataset for investigators studying samples from diseased patients.

Response: We have submitted processed data to a public repository. However, due to local regulatory policies, we are not allowed to share raw data at a massive scale. The analytical script associated with this study has also been made available via github. The availability information has been updated in the revision. Here we also provide a preview access (<https://ngdc.cnbc.ac.cn/omix/preview/p9U2RQYf>) before publication.

Major Comment 2: The conclusions are not fully supported by the data presented and should be reworded. The conclusion that microbial cfDNA is a potential biomarker of disease is accurate. Referring to the presence of cfDNA in the blood as blood microbiota risks creating confusion, as the presence of DNA does not necessarily imply the presence of intact bacterial microbiota. "Blood microbiota may represent or contribute to the first step in the kinetics of disease" is speculative and not a conclusion of the study. Considering this study is in healthy individuals, the presence of gut commensals doesn't indicate pathogens

driving disease, and the presence of viral sequence reflects chronic infections rather than the first step of acute infection.

Response: In this revised manuscript, we have rephased the conclusion to reflect such clarification, hopefully no direct confusion will be further triggered.

Minor comments 1: HHV8 is mentioned in line 237 in reference to figure 1D, but not listed in the figure.

Response: Sorry for the confusion. We hope to list various herpesviruses detected, not necessarily on the top 15 list on the figures. The low abundance state of HHV8 is clarified. We have revised this sentence.

Minor comments 2: Suggest removing the subjective descriptor of "dramatic" when describing the regional differences in Shannon's diversity (line 241). The results show the majority are 5.3 and range from 5 to 5.6. Shannon diversity is somewhat contextual and dramatic in disease conditions is used to refer to differences between >5 and <1 .

Response: Thanks for the kind comment. We have changed the subjective descriptor with "noticeable" instead of "dramatic" as your suggestion.

Minor comments 3: Line 263. This reviewer cannot find "Bacillus thuringiensis serovar" on Figure 3B that is stated as a key finding in figure.

Response: We are sorry for the typesetting mistake when rearranging the layout of Figure3. Originally, we listed top 50 bacteria in the heatmap, while unfortunately Bacillus thuringiensis serovar was left out when shortening the list, however the statement regarding its special regional enrichment was kept and unchecked. We have rearranged the image source of Figure3 to make the top 50 abundant bacteria visible.

Minor comments 4: Line 277. Can ClinicalFreq be defined so readers can properly interpret ClinFreq=36% and PopFreq=0.008 for HBV?

Response: In the revised paper, we have made it contextually clearer and added a few interpretations of the differences and similarities between data sources, showed in revised manuscript as "the ClinicalFreq is predefined elsewhere, mostly as positive rate of serological screening test among the individual study population, while PopFreq is the frequency of non-zero samples of a particular organism directly derived from the indicator matrix of this study population".

Minor comments 5: Line 294. If the authors consider the increase in the abundance of *Propionibacterium acne* to be "slight" compared with *Bifidobacterium bifidum*, this should be described further as the fold change and p values are similar on Fig. 5D.

Response: Thanks for your suggestion. Indeed, the level of change is comparable between these two bacteria. We have made it clearer by specifying the fold change (about 7.4) and P value (<0.001).

Minor comments 6: Fig. 5E. Authors should confirm that "candidate division" indicated on the figure is intended.

Response: We are sorry for the clerical confusion during manual re-annotation of the figure when adjusting font size and position. The original descriptive text in the database is "BACT_302 Candidate division TM7 single-cell isolate TM7b NZ_ABBW01000001". In this revised manuscript, we have made its original name clearer in the figure legend.

Minor comments 7: Could the authors reword line 312 "Module pink holds the largest number of viruses, mainly bacteria phages". All viruses appear pink, so it's not clear what module refers to.

Response: The original idea is that pink was used to represent virus together with its rectangular shape, whereas it was color labeled as its surrounding neighbors. The observation made for the grouping pattern of virus within the beige cluster was clarified. We revised the sentence according to your suggestion.

Minor comments 8: Two blue clusters appear in Fig 6. Consider adding "light blue" to describe the cluster in line 329, and dark blue to line 313 describing the bacterial cluster centered on *Haemophilus* phage.

Response: Thanks for your suggestion. We distinguished two blue into dark blue and light blue in revised manuscript as your suggestion.

Reviewer #2 (Public repository details (Required)):

A set of sequencing data

Reviewer #2 (Comments for the Author):

Thank you for inviting me to review the manuscript 'The peripheral blood microbiome analysis via NIPT reveals the complexity of circulating microbial cell-free DNA baselines'.

My general impression is that the described study will be of interest for the research community.

Innovative research data are described.

Major shortcomings are:

Comments 1: Title:NIPT abbreviation should be explained in the title.

Response: As your suggestion, we revised title as: the peripheral blood microbiome analysis via Noninvasive prenatal testing reveals the complexity of circulating microbial cell-free DNA baselines

Comments 2: Abstract:

The abstract is not fully clear. The aims of the study, methodology and research problems that the authors investigated are not very clear. The major findings or trends are only hinted at. Brief and more concrete description regarding approach and experimental design, materials, concrete results, conclusions and interpretation should be described.

Response: According to your kind suggestion, we have re-written the abstract and try to make it clearer.

Comments 3: Introduction:

Lines 59-63: It would be useful for the readers to understand the meaning of the term used by the authors ,baseline microbiome' and if they mean also ,core microbiome'.

Response: We think our finding might contain the “core microbiome” which is shared and detectable within the sample population subject to our experimental method, in the sense of ‘core’ is relative to ‘whole’ and almost no experimental method is free from selection bias. The adjective ‘baseline’ is to deliver the general idea of using a large population of healthy individual to establish a quantitative estimate of a test statistics. So we think the wording might not be misleading.

Comments 4: Line 63-64: 'We also investigated the bacteria and virus ecologies in human blood as well as the co-existence and interactions between these microbes.' - Results demonstrate that the study was limited to the evaluation of the microbial abundance in serum samples and that some indirect (statistical) interactions could be supposed.

Response: It is true that our study was based on re-analysis of retrospective data, which did not include any direct in vitro or in vivo experimental validation of the findings. We have amended the limitation of this study in the Discussion Section.

Comments 5: Materials:

The raw sequencing data are not publicly available. The developed and applied statistical algorithms are not available for open access.

Response: We are sorry that the availability of such massive scale sequencing data from pregnant volunteers is not allowed in China. According to the signed patient consent, these data only could only be used for the diagnostic purpose in the clinical lab by commissioned clinicians who conduct anonymous research; sharing these data to a third party is forbidden. Therefore, we are not able to provide the raw sequencing data as requested. Nonetheless, we would like to provide downstream processed data, together with analytical codes to support reproducible research. Section of “Availability of data and materials” has been updated. Here we also provide a reviewer’ link (<https://ngdc.cncb.ac.cn/omix/preview/p9U2RQYf>) before the data get public.

Comments 6: Results:

Line 308-311: Please clarify the meaning or rephrase ‘... medically utilized bacteria [1](246) and Streptomyces phage (247) as its center. Also which hypothesis you mean?’

Response: The use of Streptomyces in the manufacturing of antibiotics is well-known. We have added a reference to support this statement: Watve MG, Tickoo R, Jog MM, Bhole BD. How many antibiotics are produced by the genus Streptomyces? Arch Microbiol. 2001 Nov;176(5):386-90. doi: 10.1007/s002030100345. PMID: 11702082.

Comments 7: Figure 6. Specify the meaning of the numbers in the squares and octagons. A supplementary file would be useful.

All figures and table are appropriate.

Response: Thanks for your suggestion. The labelled number on the icon is an index of the organisms listed in the figure, with an attempt to use an ID of shorter length instead of the original long name of the organism to increase simplicity. In this revised paper, we added the annotation “the name-to-id mapping is shown in supplementary table “suppl-table1-figure6-node” in both result and figure legend.

Comments 8: Discussion:

Line 383-387: 'our current study timely established a reference range of characteristic peripheral blood microorganisms, which is helpful for early detection and interpretation of infectious diseases, especially regional epidemic diseases, and provides the pathogenic evidence for clinical diagnosis.' - This statement sounds logic but it is not supported by concrete data. It sounds like a declaration. Please rephrase.

Response: In this revised manuscript, we rephrased this sentence as “our current study timely established a reference range of characteristic peripheral blood microorganisms. Application of the data maybe

helpful for early detection and interpretation of infectious diseases, especially regional epidemic diseases, and provides the pathogenic evidence for clinical diagnosis, which need to been investigated and proved by series researches”.

Comments 9: I agree with the conclusion - 'Furthermore, these results suggested that different countries and regions may have distinct normal reference intervals of microbes in peripheral blood, which may be influenced by diet, geography, and environmental factors.' The study results give support.

Response: Thanks for your agreement. It encourages our research team a lot.

May 2, 2022

Dr. Fei Xiao
Beijing Hospital
Beijing
China

Re: Spectrum00414-22R1 (The peripheral blood microbiome analysis via NIPT reveals the complexity of circulating microbial cell-free DNA baselines)

Dear Dr. Fei Xiao:

Congratulations. I am glad to inform you that after careful consideration by both in-house and external experts, your manuscript has been accepted.

I am forwarding it to the ASM Journals Department for publication. You will be notified when your proofs are ready to be viewed.

Sincerely,

Wei-Hua Chen
Editor, Microbiology Spectrum

Journals Department
Supplemental Dataset: Accept